# Max-Margin Works while Large Margin Fails: Generalization without Uniform Convergence

**Margalit Glasgow, Colin Wei, Mary Wootters & Tengyu Ma**
Department of Computer Science
Stanford University
Stanford, CA 94305, USA
`{mglasgow,colinwei,marykw,tengyuma}@stanford.edu`

## Abstract

A major challenge in modern machine learning is theoretically understanding the generalization properties of overparameterized models. Many existing tools rely on *uniform convergence* (UC), a property that, when it holds, guarantees that the test loss will be close to the training loss, uniformly over a class of candidate models. Nagarajan & Kolter (2019b) show that in certain simple linear and neural-network settings, any uniform convergence bound will be vacuous, leaving open the question of how to prove generalization in settings where UC fails. Our main contribution is proving novel generalization bounds in two such settings, one linear, and one non-linear. We study the linear classification setting of Nagarajan & Kolter (2019b), and a quadratic ground truth function learned via a two-layer neural network in the non-linear regime. We prove a new type of margin bound showing that above a certain signal-to-noise threshold, any near-max-margin classifier will achieve almost no test loss in these two settings. Our results show that near-max-margin is important: while any model that achieves at least a $(1 - \epsilon)$-fraction of the max-margin generalizes well, a classifier achieving half of the max-margin may fail terribly. Our analysis provides insight on why memorization can coexist with generalization: we show that in this challenging regime where generalization occurs but UC fails, near-max-margin classifiers contain both some generalizable components and some overfitting components that memorize the data. The presence of the overfitting components is enough to preclude UC, but the near-extremal margin guarantees that sufficient generalizable components are present.

## 1 Introduction

A central challenge of machine learning theory is understanding the generalization of overparameterized models. While in many real-world settings deep networks achieve low test loss, their high capacity makes theoretical analysis with classical tools difficult, or sometimes impossible (Zhang et al., 2017; Nagarajan & Kolter, 2019b). Most classical theoretical tools are based on *uniform convergence* (UC), a property that, when it holds, guarantees that the test loss will be close to the training loss, uniformly over a class of candidate models. Many generalization bounds for neural networks are built on this property, e.g. Neyshabur et al. (2015; 2017b; 2018); Harvey et al. (2017); Golowich et al. (2018).

The seminal work of Nagarajan & Kolter (2019b) gives theoretical and empirical evidence that UC cannot hold in natural overparameterized linear and neural network settings. The impossibility results of Nagarajan and Kolter are strong: they rule out even UC on the smallest reasonable algorithm-dependent family of models, that is, any possible models output by learning algorithm on typical datasets. In particular, they prove that in an overparameterized linear classification problem, models found by gradient descent will achieve small test loss, but any UC bound over these models will be vacuous. In a two-layer neural network setting, Nagarajan & Kolter (2019b) empirically demonstrate the same phenomenon for the $0/1$ loss.

Many margin-based generalization bounds do not technically fit into the category of UC bounds defined by Nagarajan and Kolter, but still may be intrinsically limited for similar reasons. Classical

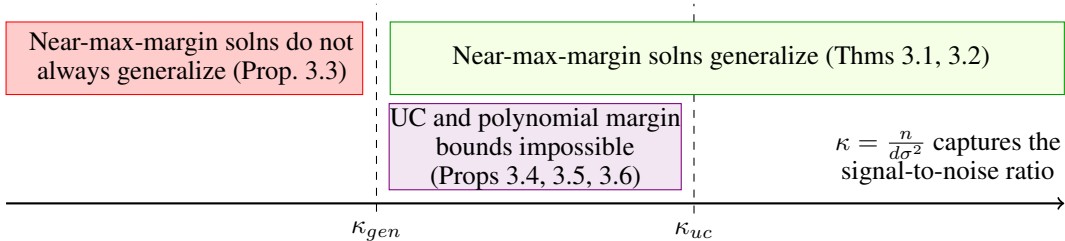

Figure 1: Thresholds for Uniform Convergence and Generalization.

margin-based generalization guarantees bounds (see eg. Shalev-Shwartz & Ben-David (2014); Kakade et al. (2009)) and related margin bounds for neural networks (Wei & Ma, 2019a; 2020; Bartlett et al., 2017; Golowich et al., 2018) scale inversely polynomially in the margin size, and are typically proved via uniform convergence on a surrogate loss (eg. the hinge loss or ramp loss) that upper bounds the $0/1$ misclassification loss. Nagarajan and Kolter's results show that any UC bound on the ramp loss is vacuous in an overparameterized linear setting, suggesting (though not proving) that classical margin bounds may not be useful. Muthukumar et al. (2021) shows empirically that such margin bounds are vacuous in a broader linear settings. In light of this, it is very important to develop theoretical tools to analyze generalization in settings where uniform convergence cannot yield meaningful bounds.

In this paper we establish novel margin-based generalization bounds in regimes where UC provably fails. These bounds guarantee generalization in the extremal case where the model has a near-maximal margin, and thus we call them *extremal margin bounds*.Indeed, near max-margin solutions are achievable by minimizing the logistic loss with weak $\ell_2$-regularization (Wei et al., 2019), and minimizing the unregularized logistic loss with gradient descent converges to a stationary points of the max-margin objective (Lyu & Li, 2019; Lyu et al., 2021). In linear settings, SGD converges to the max-margin (Nacson et al., 2019).

Our results consider two settings, the linear setting of Nagarajan & Kolter (2019b), and a commonly studied quadratic problem learned on by a two-layer neural network (Wei et al., 2019; Frei et al., 2022b). In Theorems 3.1 and 3.2, we prove that above a certain signal-to-noise threshold $\kappa_{\text{gen}}$, near-max-margin solutions will generalize. Below this threshold, max-margin solutions may not generalize (Proposition 3.3). Below a second higher threshold, $\kappa_{\text{uc}}$, uniform convergence fails (Proposition 3.4). In Figure 1 we illustrate these three regions; the main significance of our results is in the challenging middle region between $\kappa_{\text{gen}}$ and $\kappa_{\text{uc}}$ where generalization occurs, but UC fails.

Additionally in this regime where UC fails, we show that classical margin bounds can only yield loose guarantees, even for the max-margin solution (Proposition 3.5 and 3.6). We prove this by showing the existence of models that achieve a large but non-near-max-margin (e.g., half the max-margin), but do not generalize at all. This phase transition between good-margin and near-max-margin cannot be captured by classical margin bounds where the generalization guarantee decays inversely polynomially in the margin. Our extremal margin bounds are fundamentally different from classical margin bounds and are not based on uniform convergence.

Prior works have also studied the challenging regime where uniform convergence does not work. In a linear regression setting, Zhou et al. (2020) and Koehler et al. (2021) show that the test loss can be uniformly bounded for all low-norm solutions that perfectly fit the data (this uses the data-dependent interpolation condition to improve upon UC bounds); nevertheless, Yang et al. (2021) shows that such bounds are still loose on the min-norm solution. Negrea et al. (2020) suggests an alternative framework based on uniform convergence over a less complex family of *surrogate* models; they use this technique to show generalization in a linear setting and in another high-dimensional problem amenable to analysis. To our knowledge, our results are the first instance of theoretically proving generalization in a neural network setting (that is not in the NTK regime) where UC provably fails.

We leverage near-max-margins in a unified way for both the linear and nonlinear settings, and we hope that this approach will be useful more broadly in overparameterized settings. In the challenging regime of generalization without UC, good learned models contain some generalizable signal components and some overfitting components that memorize the data. Our main technique is to show that any

near max-margin solution has to contain *both* signal components and overfitting components. The overfitting component causes UC to fail, but fortunately, has a reduced influence on a random test example, whereas the signal component has a similar influence on training and test examples.

## 1.1 ADDITIONAL RELATED WORK

A large body of work highlights challenges in using classical statistical theory to explain generalization in deep learning. Experimental results (Zhang et al., 2017; Neyshabur et al., 2017a) point out that despite being large in traditional capacity measures such as Rademacher complexities, deep networks still generalize well, and new explanations are needed to understand this behavior. Belkin et al. (2018) show that similar challenges hold in kernel methods. Beyond the work of Nagarajan & Kolter (2019b), Bartlett & Long (2021) prove that in a linear interpolation setting, model-dependent generalization bounds fail for the min-norm solution. Koren et al. (2022) show that SGD can exhibit a benign underfitting phenomenon where the test loss is small but empirical loss is large.

One related body of work has focused on characterizing "benign overfitting", where the model overfits to noise in labels of the training data but still attains good test performance. Our setting differs from benign overfitting because we study models that overfit prohibitively enough to preclude UC even with *clean* data. For models that overfit to noise, (i) it still may be possible to for algorithm-dependent notions of a UC bound to explain generalization on clean data, and (ii) if the overfitting is avoided with regularization, UC bounds may also be possible. Most of the results in this area concern linear models: Bartlett et al. (2020) analyze benign overfitting in regression problems by leveraging a closed form expression for the min-norm solution. Muthukumar et al. (2021); Shamir (2022); Cao et al. (2021); Wang & Thrampoulidis (2020); Chatterji & Long (2021) and Wang et al. (2021) study classification settings. The setting of Chatterji & Long (2021) is particularly similar to ours since it considers the max-margin solution under a Gaussians mixture. The works of Muthukumar et al. (2020) and Shamir (2022) reveal that is often possible to have benign overfitting in classification, whereas in regression for the same covariate distribution, the overfitting would imply poor generalization. Also closely related to our work on linear classification is the work of Montanari et al. (2019), which asymptotically characterizes the generalization of the max-margin solution as $n, d \to \infty$. Benign overfitting in neural networks has been shown in several simple settings. Frei et al. (2022a) analyzes two-layer neural networks trained by gradient descent on linearly-separable data. Cao et al. (2022) studies benign overfitting for a two-layer simplified convolutional network. Their techniques involve decomposing the output of the network into a sum of two terms, one involving the signal feature, and one involving the noise feature. Our techniques are very different because this decomposition is not possible for a fully connected 2-layer neural network.

More broadly, a variety of new generalization bounds have been derived in hopes of explaining generalization in deep learning. While none of these bounds have been explicitly proven to succeed in regimes where UC fails, they leverage additional properties of the training data or the optimization process and thus are not directly susceptible to the critiques of Nagarajan & Kolter (2019b). Among these are works that leverage properties such as Lipschitzness of the model on the training data (Arora et al., 2018; Nagarajan & Kolter, 2019a; Wei & Ma, 2019a;b), use algorithmic stability (Mou et al., 2018; Li et al., 2019a; Chatterjee & Zielinski, 2022), or information-theoretic perspectives (Negrea et al., 2019; Haghifam et al., 2021).

Finally, a body of work seeks to draw connections between optimization and generalization in deep learning by studying implicit regularization effects of the optimization algorithm (see e.g. (Gunasekar et al., 2017; Li et al., 2017; Gunasekar et al., 2018a;b; Woodworth et al., 2020; Damian et al., 2021; HaoChen et al., 2020; Li et al., 2019b; Wei et al., 2020) and related references). Most relevent in this literature is the aforementioned work connecting gradient descent and max-margin solutions.

## 2 PRELIMINARIES

Our work achieves results in two settings. The first is a linear setting previously studied by Nagarajan & Kolter (2019b) where both the ground truth and the trained model are linear. In the second nonlinear setting, studied before by Wei et al. (2019); Frei et al. (2022b), the ground truth is quadratic, and the trained model is a two-layer neural network. In both settings, the data is drawn from a product

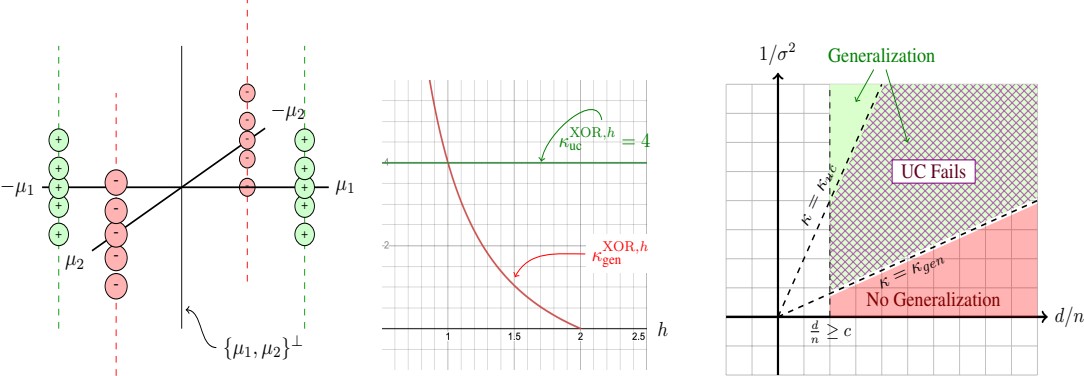

Figure 2: Left: Quadratic XOR Problem. Middle: $\kappa_{\text{gen}}^{\text{XOR},h}$ (red) and $\kappa_{\text{uc}}^{\text{XOR},h}$ (green) as a function of $h$. Right: Regions in which theorems hold. As shown in this figure, our results only hold when there is sufficient overparameterization, that is, $d \geq cn$ for a constant $c$.

distribution on features involved in the ground truth labeling function, and "junk" features orthogonal to the signal. We formalize the two settings below.

**Linear setting**

▶ **Data Distribution**. Fix some ground truth unit vector direction $\mu \in \mathbb{R}^d$. Let $x = z + \xi$, where $z \sim \text{Uniform}(\{\mu, -\mu\})$ and $\xi$ is uniform on the sphere of radius $\sqrt{d-1}\sigma$ in $d-1$ dimensions, orthogonal to the direction $\mu$. Let $y = \mu^T x$, such that $y = 1$ with probability $1/2$ and $-1$ with probability $1/2$. We denote this distribution of $(x, y)$ on $\mathbb{R}^d \times \{-1, 1\}$ by $\mathcal{D}_{\mu,\sigma,d}$.

▶ **Model.** We learn a model $w \in \mathbb{R}^d$ that predicts $\hat{y} = \text{sign}(f_w(x))$ where $f_w(x) = w^T x$.

**Setting for Two-Layer Neural Network Model with Quadratic "XOR" Ground Truth**

▶ **Data Distribution.** Fix some orthogonal ground truth unit vector directions $\mu_1$ and $\mu_2$ in $\mathbb{R}^d$. Let $x = z + \xi$, where $z \sim \text{Uniform}(\{\mu_1, -\mu_1, \mu_2, -\mu_2\})$ and $\xi$ is uniform on the sphere of radius $\sqrt{d-2}\sigma$ in $d-2$ dimensions, orthogonal to the directions $\mu_1$ and $\mu_2$. Let $y = (\mu_1^T x)^2 - (\mu_2^T x)^2$ for some orthogonal ground truth directions $\mu_1$ and $\mu_2$ (see Figure 2(left)). We denote this distribution of $(x, y)$ on $\mathbb{R}^d \times \{-1, 1\}$ by $\mathcal{D}_{\mu_1,\mu_2,\sigma,d}$. We call this the XOR problem because $y = \text{XOR}\left((\mu_1 + \mu_2)^T x, (-\mu_1 + \mu_2)^T x\right)$. For instance, if $\mu_1 = e_1$ and $\mu_2 = e_2$, then $y = x_1^2 - x_2^2$. As can be seen in Figure 2(left), this distribution is not linearly separable, and so one must use nonlinear model to learn in this setting.

▶ **Model.** Fix $a \in \{-1, 1\}^m$ so that $\sum_i a_i = 0$. The model is a two-layer neural network with $m$ hidden units and activation function $\phi$, parameterized by $W \in \mathbb{R}^{m \times d}$. $W$ (which will be learned) represents the weights of the first layer and $a$ (which is fixed) is the second layer weights. The model predicts $f_W(x) = \sum_{i=1}^m a_i \phi(w_i^T x)$, where $w_i \in \mathbb{R}^d$ denotes the $i$'th column of $W$. We work with activations $\phi$ of the form $\phi(z) = \max(0, z)^h$ for $h \in [1, 2)$, and require that $m$ is divisible by $4$[1].

We define a *problem class* of distributions to be a set of data distributions. In this paper, we work with the linear problem class $\Omega_{\sigma,d}^{\text{linear}} := \{\mathcal{D}_{\mu,\sigma,d} : \mu \in \mathbb{R}^d, \|\mu\| = 1\}$, and the quadratic problem class $\Omega_{\sigma,d}^{\text{XOR}} := \{\mathcal{D}_{\mu_1,\mu_2,\sigma,d} : \mu_1 \perp \mu_2 \in \mathbb{R}^d, \|\mu_1\| = \|\mu_2\| = 1\}$. Here $\|\cdot\|$ denotes the $\ell_2$ norm.

We will sometimes abuse notation and say that $x \sim \mathcal{D}$ instead of saying that $(x, y) \sim \mathcal{D}$.

Before proceeding, we make some comments on the parameter settings and compare to related work.

**Large dimension assumption.** In both the linear and non-linear settings, our focus is an overparameterized regime where the dimension $d$ is at least a constant factor times larger than $n$, the number of training samples. Such an assumption is mild relative to the assumptions made in related work, which require $d = \omega(n)$ (see eg. (Cao et al., 2021; Wang & Thrampoulidis, 2020; Muthukumar et al., 2021; Shamir, 2022; Chatterji & Long, 2021) on linear models; for neural networks, the closest related works of Frei et al. (2022a) and Cao et al. (2022) assume that $d \geq n^2$ or stronger). When the dimen-

---

[1]The assumption that $m$ is divisible by $4$ is for convenience, and can be removed if $m$ is large enough.

sion is sufficiently large (in particular, at least $\omega(n)$), with high probability, the max-margin solution coincides with the min-norm regression solution (see Hsu et al. (2021)), meaning the max-margin solution can be analyzed via a closed-form expression. Our work is fundamentally different from the work on linear classification which operates in the $d = \omega(n)$ regime, because in our setting when $d = \Theta(n)$, these two solutions do not coincide.

**Assumption on Data Covariance.** Many works on linear classification study more general data models which allow arbitrary decay of the eigenvalues of the covariance matrix (eg. Muthukumar et al. (2021); Wang & Thrampoulidis (2020); Cao et al. (2021)), or variance in the signal direction, that is, $x^T \mu \neq y$ (eg. Shamir (2022)). We work with a simpler distribution, which is still challenging, because it defies existing analyses built on UC or closed-form solutions.

## 2.1 BACKGROUND AND DEFINITIONS ON UNIFORM CONVERGENCE

In this subsection, we provide some definitions from Nagarajan & Kolter (2019b) on algorithm-dependent UC bounds. We also provide some definitions and background on margin bounds.

For a loss function $\mathcal{L} : \mathbb{R} \times \mathbb{R} \to \mathbb{R}$, and a hypothesis $h$ mapping from a domain $\mathcal{X}$ to $\mathbb{R}$, we define the test loss on a distribution $\mathcal{D}$ to be $\mathcal{L}_{\mathcal{D}}(h) := \mathbb{E}_{(x,y) \sim \mathcal{D}} \mathcal{L}(h(x), y)$. For a set of examples $S = \{(x_i, y_i)\}_{i \in [n]}$, we define $\mathcal{L}_S(h) := \mathbb{E}_{i \in [n]} \mathcal{L}(h(x_i), y_i)$ to be the empirical loss.

Unless otherwise specified, we will use $\mathcal{L}$ to denote the 0/1 loss, which equals 1 if and only if the signs of the two labels disagree, that is, $\mathcal{L}(y, y') = \mathbf{1}(\text{sign}(y) \neq \text{sign}(y'))$.

Typically in machine learning one considers a global hypothesis class $\mathcal{G}$ that an algorithm may explore (e.g., the set of all two-layer neural networks). A uniform convergence bound, defined below, may hold over a smaller subset $\mathcal{H}$ of $\mathcal{G}$, eg. the subset of networks with bounded norm.

**Definition 2.1** (Uniform Convergence Bound). *A uniform convergence bound with parameter $\epsilon_{\text{unif}}$ for a distribution $\mathcal{D}$, a set of hypotheses $\mathcal{H}$, and loss $\mathcal{L}$ is a bound that guarantees that*

$$\Pr_{S \sim \mathcal{D}^n} \left[ \sup_{h \in \mathcal{H}} |\mathcal{L}_{\mathcal{D}}(h) - \mathcal{L}_S(h)| \geq \epsilon_{\text{unif}} \right] \leq \frac{1}{4}. \tag{2.1}$$

A uniform convergence bound can be customized to algorithms by choosing $\mathcal{H}$ to depend on the implicit bias of an algorithm. For instance, if an algorithm $\mathcal{A}$ favors low-norm solutions, one could choose $\mathcal{H}$ to be the set of all classifiers with bounded norm. Of course, if $\mathcal{H}$ is too small, it may not be useful for proving generalization, because $\mathcal{A}$ will never output a solution in $\mathcal{H}$. We formalize the notion of choosing a useful algorithm-dependent set $\mathcal{H}$ as follows.

**Definition 2.2** (Useful Hypothesis Class). *A hypothesis class $\mathcal{H}$ is* useful *with respect to an algorithm $\mathcal{A}$ and a distribution $\mathcal{D}$ if $\Pr_{S \sim \mathcal{D}^n}[\mathcal{A}(S) \in \mathcal{H}] \geq \frac{3}{4}$.*

**Remark 2.3.** *Our definition of a uniform convergence bound on a useful hypothesis class is essentially equivalent to the definition of algorithm-dependent uniform convergence bound in Nagarajan & Kolter (2019b). We introduce new terminology since we use it later in our results on margin bounds.*

More generally, we can have generalization bounds that do not yield the same generalization guarantee for all elements of $\mathcal{H}$. Instead, their guarantee scales with some property of the hypothesis $h$ and the sample $S$. We call these *data-dependent* bounds. Such bounds are useful if the favorable property is satisfied with high probability by the algorithm of interest.

One specific type of data-dependent bound depends on the margin achieved by the classifier on the training sample. We recall the definition of a margin:

**Definition 2.4** (Margin). *The margin $\gamma(h, S)$ of a classifier $h$ on a sample $S$ equals $\min_{(x,y) \in S} y h(x)$.*

In certain parameterized hypothesis classes it is useful to define a normalized margin. If $f_W$ is $h$-homogeneous, that is, $f_{cW}(x) = c^h f_W(x)$ for a positive scalar $c$, we define the *normalized margin*

$$\bar{\gamma}(f_W, S) := \frac{\gamma(f_W, S)}{\|W\|^h} = \gamma(f_{W/\|W\|}, S), \tag{2.2}$$

where we define the norm $\|W\|$ to equal $\sqrt{\mathbb{E}_{i \in [m]}[\|w_i\|^2]}$, where $w_i$ is the $i$'th column of $W$.

We will use $\gamma^*(S)$ to denote the maximum normalized margin. When we are discussing the linear problem, we let $\gamma^*(S)$ be the max-margin over all vectors $w \in \mathbb{R}^d$ with norm 1, that is $\gamma^*(S) := \sup_{w:\|w\|_2 \leq 1} \gamma(S, f_w)$. In the XOR problem, we use $\gamma^*(S)$ to denote the max-margin over all weight matrices $W \in \mathbb{R}^{m \times d}$ with norm 1, that is $\gamma^*(S) := \sup_{W:\|W\| \leq 1} \gamma(S, f_W)$.

Most classical margin bounds prove that the generalization gap can be bounded by a term that scales inversely linearly or quadratically in the margin (Koltchinskii & Panchenko, 2002; Kakade et al., 2009). More generally, we will call margin bounds in which the generalization guarantee scales with $\left(\frac{1}{\gamma(S,f_W)}\right)^p$ for a constant $p$ a *polynomial margin bound*. Such bounds usually rely on proving uniform convergence for a continuous loss that upper bounds the $0/1$ loss. As we will show in the next section, such bounds are also intrinsically limited in regimes where UC fails on the $0/1$ loss.

In contrast to this, in our work, we prove bounds for classifiers that achieve near-maximal margins.

**Definition 2.5.** *A classifier $h$ is a $(1 - \epsilon)$-max-margin solution for $S$ if $\gamma(h, S) \geq (1 - \epsilon)\gamma^*(S)$.*

We refer to a bound that holds for $(1 - \epsilon)$-max-margin solutions as a *extremal margin bound*.

## 3 MAIN RESULTS

In the following section, we state our main results for the linear and quadratic problems, and provide intuition for our findings. As illustrated in Figure 1, and in more detail in Figure 2(right), our results show different possibilities for a near max-margin solution depending on the size of $\kappa := \frac{n}{d\sigma^2}$, a signal-to-noise parameter, where $\sigma, d$ are as in Section 2. When $\kappa$ is smaller than some threshold $\kappa_{\text{gen}}$ we are not guaranteed to have learning: even a near max-margin solution may not generalize. When $\kappa$ exceeds $\kappa_{\text{gen}}$ by an absolute constant and when $\sigma^2 \ll 1$, our results show that any near max-margin solution generalizes well. Finally, we show that if $\kappa$ is smaller than a second threshold $\kappa_{\text{uc}}$, then uniform convergence approaches will fail to guarantee generalization. All of our results additionally include an overparameterization condition that $d \geq cn$ for a constant $c$, as is pictured in Fig 2(right).

The exact thresholds $\kappa_{\text{gen}}$ and $\kappa_{\text{uc}}$ depend on the problem class of interest, but in both the linear setting and the nonlinear setting we study, we show that $\kappa_{\text{uc}} > \kappa_{\text{gen}}$. Thus we observe a regime where uniform convergence fails, but generalization still occurs for near max-margin solutions.

For the linear problem, we define the universal constants

$$\kappa_{\text{gen}}^{\text{linear}} := 0 \text{ and } \kappa_{\text{uc}}^{\text{linear}} := 1. \tag{3.1}$$

For the XOR problem with activation $\text{relu}^h$, for $h \in [1, 2)$, we define the constants

$$\kappa_{\text{gen}}^{\text{XOR},h} := \text{ the solution to } 2^{\frac{1}{h}}\sqrt{\frac{2}{\kappa}} = \sqrt{\frac{\kappa}{4 + \kappa}} + \sqrt{\frac{16}{\kappa(4 + \kappa)}} \text{ and } \kappa_{\text{uc}}^{\text{XOR},h} := 4. \tag{3.2}$$

The constants are pictured in Figure 2(right) as a function of $h$. Observe that for $h \in (1, 2)$, we have $\kappa_{\text{gen}}^{\text{XOR},h} < \kappa_{\text{uc}}^{\text{XOR},h}$, and $\kappa_{\text{gen}}^{\text{XOR},h} > 0$. When $h = 1$ and the activation is relu, we have $\kappa_{\text{gen}}^{\text{XOR},h} = \kappa_{\text{uc}}^{\text{XOR},h}$, and thus we do not expect to have a regime where uniform convergence fails, but max-margin solutions generalize. We elaborate more intuitively on why $h > 1$ allows for generalization without UC in Section A.

Our first theorem states that when $\kappa > \kappa_{\text{gen}}$, any near-max-margin solution generalizes.

**Theorem 3.1** (Extremal-Margin Generalization for Linear Problem). *Let $\delta > 0$. There exist constants $\epsilon = \epsilon(\delta)$ and $c = c(\delta)$ such that the following holds. For any $n, d, \sigma$ and $\mathcal{D} \in \Omega_{\sigma,d}^{\text{linear}}$ satisfying $\kappa_{gen}^{\text{linear}} + \delta \leq \kappa \leq \frac{1}{\delta}$, and $\frac{d}{n} \geq c$, then with probability $1 - 3e^{-n}$ over the randomness of a training set $S \sim \mathcal{D}^n$, for any $w \in \mathbb{R}^d$ that is a $(1 - \epsilon)$-max-margin solution (as in Definition 2.5), we have $\mathcal{L}_{\mathcal{D}}(f_w) \leq e^{-\frac{n}{36d\sigma^4}} + e^{-n/8}$.*

Attentive readers may observe that since $\kappa_{\text{gen}}^{\text{linear}} = 0$, Theorem 3.1 can guarantee asymptotic generalization for some sequences of parameters $(n_i, d_i, \sigma_i)_{i \geq 1}$ even when $\kappa_i = \frac{n_i}{d_i\sigma_i^2} = o_{i\to\infty}(1)$, as long as $\sigma_i^2$ decays fast enough. In Theorem C.4 in the appendix, we state a more detailed version of this theorem which states the exact dependence of $c$ and $\epsilon$ on $\delta$, yielding precise results for $\kappa = o(1)$.

We prove a similar generalization result for XOR problem learned on two-layer neural networks.

**Theorem 3.2** (Extremal-Margin Generalization for XOR on Neural Network). *Let $h \in (1, 2)$, and let $\delta > 0$. There exist constants $\epsilon = \epsilon(\delta)$ and $c = c(\delta)$ such that the following holds. For any $n, d, \sigma$ and $\mathcal{D} \in \Omega_{\sigma,d}^{\mathrm{XOR}}$ satisfying $\kappa = \frac{n}{d\sigma^2} \geq \kappa_{gen}^{\mathrm{XOR},h} + \delta$ and $\frac{d}{n} \geq c$, then with probability $1 - 3e^{-n/c}$ over the training set $S \sim \mathcal{D}^n$, for any two-layer neural network with activation function $relu^h$ and weight matrix $W$ that is a $(1 - \epsilon)$-max-margin solution (as in Definition 2.5), we have $\mathcal{L}_{\mathcal{D}}(f_W) \leq e^{-\frac{1}{c\sigma^2}}$.*

This theorem guarantees meaningful results whenever $\sigma$ is small enough. To see this, note that the assumptions of the theorem require that $\frac{d}{n} \in \left[c, \frac{1}{\sigma^2(\kappa_{\mathrm{gen}}^{\mathrm{XOR},h} + \delta)}\right]$. If $\sigma$ is small enough (in terms of $\delta$), this interval is non-empty. Further, the generalization guarantee is good if $\sigma$ is small enough (since $\exp(-1/(c\sigma^2))$ tends to 0 as $\sigma$ approaches 0). For instance consider a setting where $d \gg n$, and $\sigma^2 = \frac{n}{d}$. Then our theorem guarantees that $\mathcal{L}_{\mathcal{D}}(f_W) \ll 1$.

**Key intuitions for generalization theorems.** We demonstrate the gist of the analysis for the linear problems with some simplifications. It turns out that two special solutions merit particular attention: (i) the good solution $w_{\mathrm{g}} = \mu$ that generalizes perfectly, and (ii) the bad overfitting solution $w_{\mathrm{b}} :\approx \frac{1}{\sqrt{nd\sigma}} \sum_j y_j \xi_j$ that memorizes the "junk" dimension of the data, and satisfies $\xi_i^T w_{\mathrm{b}} \approx \frac{1}{\sqrt{nd\sigma}} y_i |\xi_i|^2 = y_i \sqrt{\frac{d\sigma^2}{n}}$ for all $i$. [2] We examine the margin of the two solutions and have

$$\bar{\gamma}(w_{\mathrm{g}}, S) = 1 \text{ and } \bar{\gamma}(w_{\mathrm{b}}, S) \approx \sqrt{\frac{d\sigma^2}{n}}. \tag{3.3}$$

At first glance, one might conclude that when $\bar{\gamma}(w_{\mathrm{g}}, S) < \bar{\gamma}(w_{\mathrm{b}}, S)$, the max margin solution will be $w_{\mathrm{b}}$, which does not generalize. However, our key observation is that any (near) max margin solution $w$ always contains a mixture of both $w_{\mathrm{g}}$ and $w_{\mathrm{b}}$. When the $w_{\mathrm{g}}$ component is small but non-trivial and the $w_{\mathrm{b}}$ component is large, the solution can simultaneously generalize but contain a large enough overfitting component to preclude UC.

More concretely, suppose we consider the margin of a linear mixture $w = \alpha w_{\mathrm{g}} + \beta w_{\mathrm{b}}$ satisfying $\alpha^2 + \beta^2 = 1$ so that $\|w\|_2 = 1$. It is easy to see that the margin on the training set is

$$\bar{\gamma}(w, S) = \alpha \bar{\gamma}(w_{\mathrm{g}}, S) + \beta \bar{\gamma}(w_{\mathrm{b}}, S) \tag{3.4}$$

Meanwhile, the margin on an test example $x$ is only slightly affected by $w_{\mathrm{b}}$:

$$\bar{\gamma}(w, x) \approx \alpha \bar{\gamma}(w_{\mathrm{g}}, S) \pm \beta w_{\mathrm{b}}^T x \approx \alpha \bar{\gamma}(w_{\mathrm{g}}, S) \pm \beta \bar{\gamma}(w_{\mathrm{b}}, S) \sqrt{\frac{n}{d}}. \tag{3.5}$$

The effect $w_{\mathrm{b}}^T x$ of the bad solution on the test sample is is smaller than $\bar{\gamma}(w, S)$ by a $\sqrt{\frac{n}{d}}$ factor because $x$ is a high dimensional random vector, and thus mostly orthogonal to $w_{\mathrm{b}}$. Therefore, even if the margin on the training set mostly stems from the bad overfitting solution, that is, $\alpha \bar{\gamma}(w_{\mathrm{g}}, S) < \beta \bar{\gamma}(w_{\mathrm{b}}, S)$, the model may still generalize as long as $\alpha \bar{\gamma}(w_{\mathrm{g}}, S) \geq \beta \bar{\gamma}(w_{\mathrm{b}}, S) \sqrt{\frac{n}{d}}$.

The optimal $\alpha, \beta$ satisfying $\alpha^2 + \beta^2 = 1$ that maximize the margin turns out to be proportional to the original margin: $\frac{\alpha}{\beta} = \frac{\bar{\gamma}(w_{\mathrm{g}}, S)}{\bar{\gamma}(w_{\mathrm{b}}, S)}$, yielding a max-margin of $\sqrt{\bar{\gamma}(w_{\mathrm{g}}, S)^2 + \bar{\gamma}(w_{\mathrm{b}}, S)^2} \approx \sqrt{\frac{d\sigma^2 + n}{n}}$.

Therefore, we have $\frac{\alpha \bar{\gamma}(w_{\mathrm{g}}, S)}{\beta \bar{\gamma}(w_{\mathrm{b}}, S)} = \frac{\bar{\gamma}(w_{\mathrm{g}}, S)^2}{\bar{\gamma}(w_{\mathrm{b}}, S)^2}$. In other words, we should expect reasonable generalization of near-max margin solutions as long as $\frac{\bar{\gamma}(w_{\mathrm{g}}, S)}{\bar{\gamma}(w_{\mathrm{b}}, S)} > (\frac{n}{d})^{1/4}$, which by eq. 3.3 occurs when $\frac{n}{d\sigma^4} \gg 1$.

In Appendix A, we describe the challenges that arise when adapting these intuitions to nonlinear setting, and our techniques for overcoming them.

Before proceeding to our lower bounds, observe that a typical margin bound for the linear setting would yield $|\mathcal{L}_{\mathcal{D}}(w) - \mathcal{L}_S(w)| \leq \frac{2|x|}{\sqrt{n}\bar{\gamma}(S,w)} \approx \frac{2\sqrt{d}\sigma\sqrt{1+1/\kappa}}{\sqrt{n}}$, which is at least 2 for $\kappa \leq \kappa_{\mathrm{uc}}^{\mathrm{linear}} = 1$.

---

[2] More precisely, we will choose $w_{\mathrm{b}}$ to be the rescaled min-norm vector satisfying $\xi_i^T w_{\mathrm{b}} = y_i$ for all $i$. This distinction is important in the case when $d$ is only a constant factor larger than $n$, and the solution $\frac{1}{\sqrt{nd\sigma}} \sum_j y_j \xi_j$ does not necessarily correctly classify the training data.

We now proceed to present our lower bounds, which show when near max-margin solutions may not always generalize, and when UC bounds and polynomial margin bounds are impossible.

If $\kappa < \kappa_{\text{gen}}$, it is possible that a near-max margin solution does not generalize at all. Since $\kappa_{\text{gen}} = 0$ in the linear setting, we only state this result for the XOR problem.

**Proposition 3.3** (Region where Max-Margin Generalization not Guaranteed). *Let $h \in (1, 2)$, and let $\epsilon > 0$. There exists a constant $c = c(\epsilon)$ such that the following holds. For any $n, d, \sigma$ and $\mathcal{D} \in \Omega_{\sigma,d}^{\text{XOR}}$ satisfying $\kappa \leq \kappa_{gen}^{\text{XOR},h} - \epsilon$ and $\frac{d}{n} \geq c$, with probability $1 - 3e^{-n/c}$ over $S \sim \mathcal{D}^n$, there exists some $W$ with $\|W\| = 1$ and $\gamma(f_W, S) \geq (1 - \epsilon)\gamma^*(S)$ such that $\mathcal{L}_\mathcal{D}(f_W) = \frac{1}{2}$.*

Theorems 3.2 and Prop. 3.3 demonstrate that in the XOR problem, there is a threshold in $\kappa$ above which generalization occurs. If $\kappa$ is above this threshold, we achieve generalization when $\sigma^2 \ll 1$.

The next proposition states that when $\kappa < \kappa_{\text{uc}}$, any algorithm-dependent uniform convergence bounds will be vacuous, that is, its generalization guarantee will be arbitrarily close to 1. We state our results for the linear and XOR neural network settings together; we state the more complicated XOR result in full and then mention how the linear result differs.

**Proposition 3.4** (UC Bounds are Vacuous). *Fix any $h \in (1, 2)$, and $\delta > 0$. For any $n, d, \sigma$ and $\mathcal{D} \in \Omega_{\sigma,d}^{\text{XOR}}$, if $\kappa_{gen}^{\text{XOR},h} + \delta \leq \kappa \leq \kappa_{uc}^{\text{XOR},h} - \delta$, there exist strictly positive constants $\epsilon = \epsilon(\delta)$ and $c = c(\delta)$ such that the following holds. Let $\mathcal{A}$ be any algorithm that outputs a $(1 - \epsilon)$-max-margin two-layer neural network $f_W$ for any $S \in (\mathbb{R}^d \times \{1, -1\})^n$. Let $\mathcal{H}$ be any hypothesis class that is useful for $\mathcal{D}$ (as in Definition 2.2). Suppose that $\epsilon_{\text{unif}}$ is a uniform convergence bound for $\mathcal{D}$ and $\mathcal{H}$ that is, $\Pr_{S \sim \mathcal{D}^n}[\sup_{h \in \mathcal{H}} |\mathcal{L}_\mathcal{D}(h) - \mathcal{L}_S(h)| \geq \epsilon_{\text{unif}}] \leq 1/4$. Then if $\frac{d}{n} \geq c$ and $n > c$, we must have $\epsilon_{\text{unif}} \geq 1 - \delta$.*

*A similar result holds for the linear problem with $\kappa_{gen}^{\text{linear}} + \delta < \kappa < \kappa_{uc}^{\text{linear}} - \delta$ and any $\mathcal{D} \in \Omega_{\sigma,d}^{\text{linear}}$. In this case we achieve the guarantee that $\epsilon_{\text{unif}} \geq 1 - e^{-\frac{n}{36d\sigma^2}} - e^{-n/8}$.*

Prop. 3.4 is proved using the same technique as in Nagarajan & Kolter (2019b): we show that with high probability over $S \sim \mathcal{D}^m$, the hypothesis $\mathcal{A}(S)$ has good generalization, but on an "oppositite" dataset $\psi(S)$ with the junk components reversed, the empirical error of $\mathcal{A}(S)$ is close to 1. This large gap between empirical error and generalization forces $\epsilon_{\text{unif-alg}}$ to be large.

Further extending this technique, we can also show the limitations of classical polynomial margin bounds which achieve an bound that scales inversely polynomially with $\gamma(h, S)$. We show that with high probability over $S \sim \mathcal{D}^m$, the hypothesis $\mathcal{A}(\psi(S))$ has a large margin on the set $S$ (a constant fraction times the max-margin), but poor generalization on $\mathcal{D}$. Since any polynomial margin bound cannot predict much better generalization for the max-margin solution than for a solution with a constant-fraction of the max-margin, we conclude that any such margin bound is far from showing good generalization for the max-margin solution.

One subtlety to this approach is that here (unlike in the work of Nagarajan & Kolter (2019b)), the "opposite" data set $\psi(S)$ is defined to be the data set with the signal features reversed. Thus we can only show the limitations of polynomial margin bounds that are useful for *both* $\mathcal{D}$ and for its "oppostite" distribution $\psi(\mathcal{D})$, which has the opposite ground-truth vector(s), which is a slightly stronger assumption than in the work of Nagarajan & Kolter (2019b). [3] Formally, if $\mathcal{D} = \mathcal{D}_{\mu,\sigma}^{\text{linear}}$, then we define $\psi(\mathcal{D}) := \mathcal{D}_{-\mu,\sigma}^{\text{linear}}$. If $\mathcal{D} = \mathcal{D}_{\mu_1,\mu_2,\sigma}^{\text{XOR}}$, then $\psi(\mathcal{D}) := \mathcal{D}_{\mu_2,\mu_1,\sigma}^{\text{XOR}}$.

The following results state that if $\kappa < \kappa_{\text{uc}}$, then certain types of margin bounds cannot yield better than constant test loss on even the max-margin solution.

**Proposition 3.5** (Polynomial Margin Bounds Fail for Linear Problem). *Fix $\delta > 0$. For any $n, d, \sigma$ and $\mathcal{D} \in \Omega_{\sigma,d}^{\text{linear}}$ such that $\kappa_{gen}^{\text{linear}} + \delta < \kappa < \kappa_{uc}^{\text{linear}} - \delta$ and $\frac{d}{n} \geq c$, the following holds. Let $\mathcal{A}$ be any algorithm so that $\mathcal{A}(S)$ outputs a $(1 - \epsilon)$-max-margin solution $f_w$ for any $S \in (\mathbb{R}^d \times \{1, -1\})^n$.*

---

[3]We believe considering such types of margin bounds is natural. Indeed, for most problems, the designer of the generalization bound would not know in advance the ground truth distribution, but might know that the data comes from some problem class, e.g., linearly separable distributions, or linearly separable distributions with a sparse ground truth vector. In such cases, they would likely have to design a generalization bound that holds for data coming from two distributions with opposite ground truths.

If we make this same stronger two-distribution assumption in Prop. 3.4, we can additionally rule out *one-sided* UC bounds, which only upper bound $\mathcal{L}_\mathcal{D} - \mathcal{L}_S$.

*Let $\mathcal{H}$ be any hypothesis class that is useful for $\mathcal{A}$ (as in Definition 2.2) on both $\mathcal{D}_{\mu,\sigma}^{\text{linear}}$ and $\mathcal{D}_{-\mu,\sigma}^{\text{linear}}$. Suppose that there exists an polynomial margin bound of integer degree $p$: that is, there is some $G$ that satisfies for $\tilde{D} \in \{\mathcal{D}, \psi(\mathcal{D})\}$,*

$$\Pr_{S \sim \mathcal{D}^n}\left[\sup_{h \in \mathcal{H}} \mathcal{L}_{\tilde{\mathcal{D}}}(h) - \mathcal{L}_S(h) \geq \frac{G}{\gamma(h, S)^p}\right] \leq \frac{1}{4}.$$

*Then with probability $\frac{1}{2} - 3e^{-n}$ over $S \sim \mathcal{D}^n$, the margin bound is weak even on the max-margin solution, that is, $\frac{G}{\gamma^*(S)^p} \geq \max\left(\frac{1}{c}, 1 - e^{-\frac{\kappa}{36\sigma^2}} - e^{-n/8} - \frac{3\kappa}{c}\right)^p$, which is more than an absolute constant.*

This theorem says that no polynomial margin bound will be able to show that the test error of the max-margin solution is less than an absolute constant. We know however from Theorem 3.1 that in this same regime, the test error of the max-margin solution can be arbitrarily small for small enough $\sigma$. Thus no polynomial margin bound can predict this behaviour.

The attentive reader again may notice that if $\kappa \to 0$ as $n$ and $d$ grow, but generalization occurs, any such margin bound is vacuous, in that $\frac{G}{\gamma^*(S)^p} \to 1$. In Prop **??**, we prove a more precise version, yielding the exact dependence of $c$ and $\epsilon$ on the gap between $\kappa$ and the boundaries $\kappa_{\text{uc}}^{\text{linear}}$ and $\kappa_{\text{gen}}^{\text{linear}}$.

We achieve a similar result in the XOR setting.

**Proposition 3.6** (Polynomial Margin Bounds Fail for XOR on Neural Network). *Fix an integer $p \geq 1$, and any $\epsilon > 0$. There exists $c = c(p, \epsilon)$ such that the following holds for any $n, d, \sigma$ and $\mathcal{D} \in \Omega_{\sigma,d}^{\text{XOR}}$ with $\kappa_{\text{gen}}^{\text{XOR},h} + \epsilon < \kappa < \kappa_{\text{uc}}^{\text{XOR},h} - \epsilon$, $\frac{d}{n} \geq c$ and $n \geq c$. Let $\mathcal{H}$ be any hypothesis class such that for $\tilde{\mathcal{D}} \in \{\mathcal{D}, \psi(\mathcal{D})\}$,*

$$\Pr_{S \sim \tilde{\mathcal{D}}^n}\left[all\ (1 - \epsilon)\text{-max-margin two-layer neural networks } f_W \text{ for } S \text{ lie in } \mathcal{H}\right] \geq 3/4.$$

*Suppose that there exists an polynomial margin bound of degree $p$: that is, there is some $G$ that satisfies for $\tilde{\mathcal{D}} \in \{\mathcal{D}, \psi(\mathcal{D})\}$,*

$$\Pr_{S \sim \tilde{\mathcal{D}}^n}\left[\sup_{h \in \mathcal{H}} \mathcal{L}_{\tilde{\mathcal{D}}}(h) - \mathcal{L}_S(h) \geq \frac{G}{\gamma(h, S)^p}\right] \leq \frac{1}{4}.$$

*Then with probability $\frac{1}{2} - 3e^{-n/c}$ over $S \sim \mathcal{D}^n$, on the max-margin solution, the generalization guarantee is no better than $\frac{1}{c}$, that is, $\frac{G}{\gamma^*(S)^p} \geq \frac{1}{c}$.*

**Remark 3.7.** *The polynomial margin impossibility results is slightly weaker for the XOR problem. Namely, the hypothesis class $\mathcal{H}$ we consider is larger in the XOR problem: it must contain with probability $\frac{3}{4}$ any near max-margin solution, instead of just the one output by $\mathcal{A}$.*

The combination of our generalization results and our margin possibility results suggest a phase transition in how the margin size affects generalization. If the margin is near-maximal, Theorems 3.1 and Prop. 3.2 show that we achieve generalization. Meanwhile, the proof of Props 3.5 and 3.6 suggest that solutions achieving a constant factor of the maximum margin may not generalize.

The proofs of all of our results concerning the linear problem are given in Section C. The proofs for the XOR problem are in Section D.

## 4 CONCLUSION

In the work, we give novel generalization bounds in settings where uniform convergence provably fails. We use a unified approach of leveraging the extremal margin in both a linear classification setting and a non-linear two-layer neural network setting. Our work provides insight on why memorization can coexist with generalization.

Going beyond our results, it is important to find broader tools for understanding the regime near the boundary of generalization and no generalization. We conclude with several concrete open directions in this vein. One question is how to prove generalization without UC when $d < n$, but the model itself (e.g. a neural network) is overparameterized, and thus can still overfit to the point of UC failing. A second direction asks if we can prove similar results in the non-linear network setting for the solution found by gradient descent, if this solution is not a near max-margin solution. Indeed, in a non-convex landscape, it not guaranteed that that gradient descent will find the max-margin solution.

## ACKNOWLEDGMENTS

We thank Jason Lee for helpful discussions. MW acknowledges the support of NSF Grant CCF-1844628 and a Sloan Research Fellowship. TM is supported by NSF IIS 2045685.

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
