# OpenReview forum: "Max-Margin Works while Large Margin Fails: Generalization without Uniform Convergence"
_ICLR.cc/2023/Conference — ICLR 2023 poster_

### Official Review · Reviewer_qDFy · 2022-10-23

**Confidence:** 3
**Clarity, Quality, Novelty And Reproducibility:** Paper is well-written and the present…
**Correctness:** 3
**Technical Novelty And Significance:** 3
**Empirical Novelty And Significance:** Not applicable
**Recommendation:** 6

**Strength And Weaknesses:**

Strength:

- The new generalization bound developed in this paper is in contrast to the existing uniform convergence bounds, which have been shown to be vacuous in some settings.

- The main discovery of this paper is to show that when the SNR of the data is in a certain range, uniform convergence bounds fail but near-max-margin solutions generalize. In this case, near-max-margin classifiers contain some generalizable components and some overfitting components that memorize the data.

Question:

- The data samples are generated by perturbing two points with random noises. What would happen if the data samples from both classes are overlapping and do not have a clear margin?

**Summary Of The Paper:**

This paper develops a new type of generalization bound for both the linear classification setting and the quadratic ground truth function learned via a two-layer neural network in the non-linear setting. The main result of this paper is to show that above a certain signal-to-noise threshold, any near-max-margin classifier will generalize, while in a certain regime of the signal-to-noise threshold, uniform convergence generalization bounds are vacuous but the new generalization bound shows that memorization can coexist with generalization.

**Summary Of The Review:**

See above

---

> ### Author Response · Authors · 2022-11-11
> **Response to Review**
>
> We thank the reviewer for their comments. We are glad they find our results to be novel and to provide generalization results in areas where existing tools do not work. Given that the review finds our results to be quite significant, we are unsure what the reviewer finds to be the main weaknesses of the paper. If the review has any additional concerns, we would be happy to discuss.
>
> The reviewer’s question about what would happen if there were overlapping data. In this case, because we are in an overparameterized setting, the data would still be separable with a large margin. Even if there were no signal at all, the max-margin solution would have a margin at least a constant factor as large as the setting with signal. This is because, as explained in our “Key Intuitions”, the overfitting component can explain more than half of the margin. If we considered a setting with a constant factor of label noise, we believe similar techniques could be used to show that the max-margin solution generalizes.

---

### Official Review · Reviewer_pEEx · 2022-10-23

**Confidence:** 4
**Correctness:** 4
**Technical Novelty And Significance:** 4
**Empirical Novelty And Significance:** 3
**Recommendation:** 8

**Clarity, Quality, Novelty And Reproducibility:**

This paper extends the existing papers. The techniques in this paper are somehow novel.
Besides, this paper is well-written and easy to follow.

**Strength And Weaknesses:**

Strength:
1. This paper focuses on a very important topic: generalization.
2. This paper extends the previous results on linear cases to non-linear cases.
3. Further, the authors also propose two thresholds for the signal-to-noise ratio. Under the first threshold, UC fails, and under the second threshold, generalization fails. The two thresholds are proven to be tight.
4. The authors further provide some insights on the notion of margin, which explains how generalization works while UC fails.

Flaws:
1. I am not sure how many technical contributions are contained in this paper. The authors could restate the theoretical contributions more carefully. For example, from the linear case to the non-linear case, which technique is the core? And how does the non-linear case differ from the other non-linear cases since the XOR case proposed in this paper is solvable?
2. It would be more interesting if the authors could provide some difficulties when generalizing the results to general cases, which, of course, is pretty hard.
3. Although the title claims "max margin" and "small margin", this is only discussed in the final part of the paper and has a weak relationship with the main text.

**Summary Of The Paper:**

This paper is an extension of the existing paper by Nagarajan & Kolter.
Previously, Nagarajan & Kolter show that there exist cases such that the true generalization gap is non-vacuous while uniform convergence fails, using an overparameterized linear regression case.
In this paper, the authors show that such a phenomenon happens in both linear and non-linear cases and provides the threshold for the phenomenon happens.
Overall, I think this paper is very interesting and extends the scope of generalization analysis.

**Summary Of The Review:**

Although there are some small flaws in this paper, I believe this is an interesting topic and brings some insights into the deep learning community.
Therefore, I give a general rating of acceptence.

---

> ### Author Response · Authors · 2022-11-11
> **Response to Review**
>
> We thank the reviewer for their positive feedback! We are glad they find the result to be very interesting and to provide new insights on generalization of max-margin solutions, particularly in non-linear settings. We respond to the reviewers feedback below.
>
>   - **Technical Contributions (Novelty of Techniques).** This is a great question. Due to space limitations in the main body, we have included a detailed proof sketch in Appendix A, which contains the main technical difficulties, intuitions, and insights for both the linear and non-linear setting. As described in this section, the “core” of the technique, which unifies both linear and non-linear problems, is that we decompose the near-max-margin solution into “overfitting” and “signal” components, and show that (a) the overfitting component does not affect test samples much and (b) any near-max margin solution must a “signal” component which stands well alone. Appendix A then describes why showing (b) is challenging in the non-linear setting, and explains one of our key technical insights: how (b) can be proved by reducing the problem of max-margin maximization over a 2-layer neural network to an 3-variable optimization problem. Intuitively, this 3-variable optimization problem (Def A.3) concerns how much a single neural should trade off between 3 quantities: “b”: the strength of the signal component, “c”: the strength of a overfitting component pointing towards the rightmost cluster (see Fig 2), and “d”: the strength of a overfitting component pointing towards the leftmost cluster. These 3 variables are the analogs of $\alpha$ and $\beta$, the proportions of the “good” and “bad” solutions in the linear problem.
>
>
>     We have revised our Key Intuitions in the main body to point to this Appendix, and highlight this technical contribution for neural networks. While unfortunately we do not have space to include this discussion in the main body, we will also further emphasize the connections between the linear and non-linear setting in Appendix A.
>
>
>     Could the reviewer clarify what they mean by “how does the non-linear case differ from the other non-linear cases since the XOR case proposed in this paper is solvable?”.
>   - **Challenges of generalizing the results.** In general, we believe our results and techniques could be adapted to other non-linear settings with well-separated data where the dimension of the ambient data d is greater than n, the number of samples. Doing so would require modifying the 3-variable optimization problem in Def A.3.
>
>     The main difficulty of our techniques when $n > d$ is that there will be a lot of diversity among the directions of the $m$ neurons. This behavior cannot be reduced to a finite-variable optimization problem as described above.
>   - **Discussion of “Large Margin”.** We do discuss the difference between the max-margin and large-margin solutions in our introduction, in the paragraph beginning “Additionally in this regime where UC fails”. Here we describe that there exist large-margin solutions that do not generalize at all. As the reviewer notes, this distinction is described in more detail at the end of the paper.

---

### Official Review · Reviewer_TxXm · 2022-10-26

**Confidence:** 4
**Correctness:** 4
**Technical Novelty And Significance:** 3
**Empirical Novelty And Significance:** 2
**Recommendation:** 6

**Clarity, Quality, Novelty And Reproducibility:**

The clarity and quality are generally good. In terms of novelty, the authors may need to add more discussions on the comparison to prior works, as commented in the weakness section.


**Strength And Weaknesses:**


Strength:
* This paper provides a new margin-based generalization bound for linear and two-layer network models on certain data set.
* This paper shows that the max-margin solution will contain both generalizable and overfitting components.
* This paper clearly identifies the settings that the margin-based results and uniform convergence based results will give vanishing or vacuous generalization bound.

Weakness:
* The generalization analysis in linear case is quite similar to that in [Chatterji and Long, 2021], a thorough discussion should be added. In fact, [Chatterji and Long, 2021] study the generalization error of gradient descent for learning high-dimensional Gaussian mixture model (their dimension assumption is a bit stronger than this paper as they require $d>n^2\log(n)$), which can be also extended to derive certain margin-based generalization results.

Chatterji, N. S., & Long, P. M. (2021). Finite-sample Analysis of Interpolating Linear Classifiers in the Overparameterized Regime. J. Mach. Learn. Res., 22, 129-1.
* As claimed by the authors, the max-margin solution will contain both generalizable and nongeneralizable (or overfitting) components, then why should one still prefer to consider max-margin solution? I can understand this for linear case since GD will finally give max-margin solution, but for nonlinear case, I have two concerns: 1) is (2.3) a good definition of the margin; and 2) why is studying such a max-margin solution, or near max-margin solution a good direction to understand the generalization?
* In Proposition 3.3, the failure of near-max-margin solution will not generalization is only shown in terms of existence, while it is not clear whether this “bad near-max-margin” solution is able to be discovered by training algorithms. For instance, assuming the gradient descent will finally converge to the max-margin solution, then applying proper early-stopping will likely give a  near-max-margin solution. Then will this type of near-max-margin solution give bad generalization?
* For xor case, it would be better to also consider $h\ge 2$, as studied by [Cao et al., 2021] (though the over-parameterization conditions are different). Besides, a discussion in terms of the theoretical results compared with this prior should be added (after the theorem).


**Summary Of The Paper:**

This paper is motivated by the failure of uniform convergence based generalization bound. Accordingly, this paper proves a new type of margin based generalization bound in two settings: linear and two-layer nonlinear network model. In particular, this paper provides a threshold on the signal-to-noise ratio to indicate under which regime the margin based results and uniform convergence based results will success or fail. The authors also show that the max-margin solution will contain both generalizable components and some overfitting components.


**Summary Of The Review:**

Overall this paper is well written. More discussions should be added according to my comments in the weakness section.

---

> ### Author Response · Authors · 2022-11-11
> **Response to Review**
>
> We thank the reviewer for their insightful comments! We are glad the reviewer finds our results and paper to be novel and clear. We address their comments below:
>
>   - **Comparison to Chatterji and Long.** We agree that this is relevant since they consider the max-margin solution in a Gaussian mixture model. We have revised to include discussion in the related work and where we discuss other results on linear classification. We note that other results mentioned on linear classification (eg. Shamir et al 2022) also yield results for a Gaussian mixture model. All of these analyses share some similar attributes to ours, because they leverage the fact that the component in the spiked direction(s) doesn't need to have the same norm as the ground truth: it suffices to learn a classifier whose projection onto the ground truth classifier is substantial enough.
>   - **Why do we consider max-margin?** As stated in the introduction, if a neural network with homogeneous activations is trained with gradient descent on the logistic loss, then it will converge to a stationary point of the maximum-margin objective (Lyu & Li, 2019; Lyu et al., 2021). Due to the non-convexity of the landscape, we cannot guarantee that GD or SGD will find a global maximum (that is, the max-margin solution). However, due to the difficulty of analyzing GD directly, we believe this is a very good proxy for theory, and has been considered in other recent works [eg. Ji and Telgarsky 2018, Lyu et al. 2021, Frei et al 2022]. Further, even though the max-margin contains some overfitting components, we do not believe this means it is a bad solution. In the linear setting, the GD solution will also include overfitting components, even if regularized. Thus this overfitting is unavoidable, and one of the goals of this paper is to show that it is benign.
>   - **Is a bad-near max-margin solution discovered by algorithms when $\kappa < \kappa_{gen}$?** While the reviewer is correct that the current proof only guarantees the existence of some bad near-max-margin solution, we hypothesize that a similar technique could show that any near-max-margin solution does not generalize.
>
>     In more detail, the crux of the proof of Proposition 3.3 is Lemma D.15, which shows that when $\kappa < \kappa_{gen}$, the solution to a certain 3-variable optimization problem (see Definition A.3) has $b = 0$. Here $b$ represents the component of a single neuron in the generalizable directions (see the overview in Appendix A). Thus Lemma D.15 says in the max-margin solution, most neurons will have no component in the generalizable directions. By a continuity argument, for any near-max-margin solution, most neurons will only have very small components in the generalizable directions. To show that any near-max-margin is bad, we would then have to show that the overfitting component has a sufficiently large influence on a random sample, which does not directly follow from our proofs, but we hypothesize to be true.
>
>   - **Consider h > 2 for XOR akin to Cao et al. 2022?** We have revised to include a more detailed discussion in the Related Work in comparison to the results of Cao et al. 2022. Because of the differences between their setting and ours (both of the ground truth data and neural network model), we do not think such a comparison belongs directly next to our theorem.
>
>     One key difference between our results and theirs is that Cao et al. consider a certain 2-layer CNN architecture, where the output function can be decomposed as a sum of two separate functions, each applied to half of the input vector, corresponding to the “signal” feature and the “noise” feature. This makes their architecture amenable to techniques which would not work in our fully-connected 2-layer neural network setting. Overcoming this obstacle regarding the entanglement of the signal and noise in our setting is described in detail in Appendix A (see bullet 1. on page 14), and is a key technical insight of our proofs.
>
>     For technical reasons in our proofs, we do not consider $h > 2$. The barrier is purely a function of the way we analyze the 3-variable optimization program (see Definition A.3). We expect that another analysis of this program could replace our Lemma D.15 for $h > 2$, and then our results would follow.
>
>
> Frei et al. 2022 https://arxiv.org/abs/2210.07082
>
> Ji and Telgarsky 2018. https://arxiv.org/abs/1810.02032
>
> Lyu et al. 2021 https://arxiv.org/pdf/2110.13905.pdf
>
> Shamir et al 2022 https://arxiv.org/abs/2201.11489

---

### Official Review · Reviewer_DBpB · 2022-10-27

**Confidence:** 4
**Correctness:** 3
**Technical Novelty And Significance:** 3
**Empirical Novelty And Significance:** Not applicable
**Recommendation:** 6

**Clarity, Quality, Novelty And Reproducibility:**

**Regarding Novelty**

The Theorem 3.4 on the vacuity of uniform convergence bounds is similar to a prior result from Nagarajan & Kolter (NeurIPS 2019) and uses the same proof technique. Theorems 3.5 and 3.6 on the vacuity of polynomial margin bounds are novel as far as I am aware. While equally being based on the proof technique from Nagarajan & Kolter (NeurIPS 2019), this is a key contribution, particularly in combination with Theorems 3.1 and 3.2 which show that the maximum margin classifier still generalizes.

However, the differences between this paper and Nagarajan & Kolter (NeurIPS 2019) are not obvious and demand a proper comparison.
1) Nagarajan & Kolter (NeurIPS 2019) already consider UC for the ramp loss, i.e. a polynomial margin loss.
2) According to the paper, the definition of usefulness is "essentially equivalent" to the definition of algorithm-dependent uniform convergence bounds from Nagarajan & Kolter. Yet, usefulness seems to be necessary for the results on margin bounds, i.e. algorithm-dependent uniform convergence cannot be used there.  This suggests that usefulness is not equivalent, but a stronger assumption.
3) Nagarajan & Kolter (NeurIPS 2019) argue that "showing failure of uniform convergence in these [linear] models is, in a sense, the most interesting." I would like to hear the authors perspective on this, as they do consider two layer networks specifically.
4) The third paragraph of the introduction is confusing. It begins with that the results of Nagarajan & Kolter do not apply to many margin-based generalization bounds but later it states that the results suggest that "classical margin bounds may not be useful".

The intuitions section it shows, that the max margin classifier corresponds to the empirical mean of the data. Since Theorem 3.1 assumes that the data distribution is from a family parametrized by $\mu$, is achieved, if the parameter $\mu$ is estimated accurately enough. This suggests that Theorem 3.1 could be considered as a classic mean estimation problem and the relation to the max-margin might only be by chance (or construction). I would like to see a discussion.
This observation also suggests a more direct proof; see the end of the review.

--------------------

**Regarding Clarity**

- The paper is difficult to understand and the results in Section 3 are presented quite technically. If theorem statements could be changed so that they focus on the key findings, this would be a major improvement.

- The main assumption seems to be the value of the "signal to noise ratio" $\kappa$, as it determines the generalization regime (see Fig. 1). However, in the theorem statements, there are additional implicit assumptions on $\kappa$ (e.g. Theorem 3.4 requires $\frac 1 {\kappa \sigma^2} = d/n\ge c$, Theorem 3.1 requires $d/n\ge c \frac 1 {\kappa^2}$, which is equivalent to $\kappa\ge c \sigma^2$). Similarly, the section "key intuitions for generalization theorems" requires $\frac n {d \sigma^4} = \frac \kappa {\sigma^2}$ to be large.
It seems that $\kappa$ alone is insufficient to determine the generalization behavior and Figure 1 is too simple.

- Connected to the previous point is that the assumptions of the theorems are not clear.
Many assumptions are interconnected and identifying the validity region of the theorems is difficult.
For instance, Proposition 3.5 requires $\frac n d  \le \frac 1 c \kappa^2(\kappa_{uc}-\kappa)^4$ or equivalently $\sigma^2 \le \frac 1 c \kappa(\kappa_{uc}-\kappa)^4$. But $\kappa$ is again a function of $\sigma$, so this inequality actually determines the range for $\sigma$ given $n/d$.

- The remark after Theorem 3.2 indicates that for a given probability distribution, i.e. given $\mu, \sigma, d$, the theorem only holds if the sample size $n$ is **small enough**. This is quite unusual.

- The theorems are not limited to the maximum margin classifiers but consider also ($1-\epsilon$)-maximum margin classifiers. I might be missing something, but this does not appear to be a strengthening of the results. If the bounds hold for maximum margin classifiers, then continuity implies that the bounds still hold in a sufficiently small neighborhood, e.g., in a sublevel set of the margin operator.

-----------------------------

**Regarding Reproducibility**

The proofs are kept short and many steps must be filled in by the reader. Due to the short reviewing period, I could only check proofs for the linear setting.

Below, I will list points, where I am not sure, if the proof is correct and I just missed something, or if there is a mistake. And if so, if it can be corrected such that the theorems still hold.


- **Proof of Lemma B.1:** Why does invertibility of $\Xi^T \Xi$ imply $v = \Xi (\Xi^T \Xi)^{-1} c$? By assumption $\Xi^T v = c$ and so $v= (\Xi \Xi^T)^{-1} (\Xi \Xi^T) v = (\Xi \Xi^T)^{-1} \Xi c$. Further, why does this imply that  $\|v\|$ is in the interval specified in Eq. B.5?

- **Proof of Lemma C.1:**  How is Lemma B.1 applied here?
What would be $c$ and what would be the corresponding minimum vector $v$?

- **Proof of Lemma C.3:** Typo. In Eq. C.17, "$\le$" needs to be replaced with "$\ge$".

- **Proof of Lemma A.1:** Could you elaborate on Eq C.36. I understand that $f_v^2$ is beta(1/2, (d-2)/2) distributed, but then I am out.

- **Proof of Theorem 3.1:** The last inequality uses $d>n$, but this is not listed as an assumption.

------------------

Unrelated to score. A more direct proof of Theorem 3.1 (and perhaps 3.2) seems possible. In the intuitions section it is shown, that the max margin classifier corresponds to the empirical mean of the data and that its "overfitting component" $v$ is the empirical mean of the spherical noise. Thus, concentration inequalities, e.g. vector Bernstein, allow to bound $\|| v\||$ with high probability and could be used instead of Lemma C.2.

Typo on page 7 $\sqrt{\gamma(w_b,S)^2+  \gamma(w_b,S)^2}$ change to $\sqrt{\gamma(w_g,S)^2+  \gamma(w_b,S)^2}$

**Strength And Weaknesses:**

Strengths
- The paper extends prior results by Nagarajan & Kolter (NeurIPS 2019), as it shows that not only uniform convergence but also polynomial margin bounds, might be unable to explain generalization in deep learning.
- The existence of regimes where different generalization bounds apply is very interesting and to some extent unexpected.
- In addition to linearly separable data, a variant of the XOR problem is analyzed.

Weaknesses
- Section 3 of the paper (which presents the results) is **very** difficult to understand and technical. A careful and extensive polish is necessary. This includes a more thorough comparison with related work.
- Reproducibility: The proofs are kept rather short. In consequence, I am unable to confirm the correctness of the results.

**Summary Of The Paper:**

The paper considers generalization aspects of learning algorithms on toy data. For the considered data distributions uniform convergence generalization bounds, as well as polynomial margin bounds, provably fail. Nevertheless, a maximum margin classifier can still generalize on such distributions.
This is a pure theory paper. The contributions are the quantification of the aforementioned effects, together with their proofs and discussions.

**Summary Of The Review:**

The observation that there exists data sets where max-margin classifiers work but polynomial margin bounds fail is quite interesting. This will be a good paper, if presentation is improved, i.e. theorems simplified and and put into context. However, doing so might require a major revision of the paper.

---

> ### Author Response · Authors · 2022-11-11
> **Response to Review (Part 1)**
>
> We thank the reviewer for their close reading and insightful comments. We are glad that the reviewer finds out generalization results to be very surprising (and unexpected!). The reviewer’s primary concern was the clarity of our main theorems in Section 3. The reviewer’s concerns about reproducibility (proofs) appear to be minor typos (which we have addressed) or misunderstandings. We have taken the clarity concerns very seriously, and have revised with a new figure in Figure 2(right) (more detailed than Figure 1), and a simplification of the assumptions in several of the more-complicated theorems. We believe this revision should make the regions where our results apply much more clear, and we hope that if the reviewer is satisfied by our revisions and explanations, they will increase their score.
>
> We include detailed comments below on all of the reviewer’s concerns.
>
> **Novelty**
> The main contribution of our work are our upper bounds (Thrm 3.1 and 3.2), particularly 3.2 which concerns neural networks learning a nonlinear ground truth.
> As correctly pointed out by the reviewer, Prop. 3.4 is very similar to Nagarajan and Kolter's result. Our assumptions are analogous (although we additionally consider a neural network setting) and we use the same proof approach. The main objective of this proposition is not novelty, but rather to justify that Theorem 3.1 and 3.2 yield results in a challenging regime where existing UC-based tools would fail.
> Prop 3.5 and 3.6 are novel, and while the proofs include some new ideas, they build upon the ideas of Nagarajan and Kolter. Again the main objective of this proposition is not novelty, but to further justify why the regime of 3.1 and 3.2 is challenging, and to build upon the results of NK to not only rule out UC, but also margin bounds.
>
>   - **Paragraph 3 of introduction confusing; Nagarajan and Kolter consider ramp loss.** As we explain, Nagarajan and Kolter prove that any UC bound on the ramp loss will be vacuous. Since showing UC of the ramp loss is a frequently used tool for proving classical margin bounds, this hints that "classical margin bounds may not be useful". However, their work does not suffice to prove that margin bounds will fail (eg. it could be possible to prove a polynomial margin bound by considering different losses). Further, we see no way to directly rule out classical margin bounds directly from their result (we are happy to elaborate more on this if it’s useful). Our work provably rules out any such margin bounds. We will revise to clarify this in the introduction.
>
>   - **Isn’t failure of uniform convergence in these [linear] models most interesting?** We agree that lower bounds (failure of UC) are most interesting in simple [for example linear] settings. However, the main focus of our paper is showing upper bounds (proving generalization), which are far more challenging to prove in settings with a nonlinear ground truth and a NN model. Prior to our work, we are not aware of any work showing generalization for fully connected neural networks in a nonlinear setting and beyond NTK where UC probably fails. We do indeed also prove that the failure of UC also occurs in the neural network setting, in order to justify that our new techniques are necessary.
>
>   - **Is usefulness a stronger assumption than Najaragan and Kolter’s assumption?** Usefulness for a specific distribution D is equivalent to NK’s assumption. This is the assumption we use in Proposition 3.4, our lower bound for uniform convergence of the 0/1 loss.
> In Proposition 3.5 & 3.6, we require a stronger assumption: that the [algorithm-dependent] hypothesis class is useful for TWO distributions: D and $\psi(D)$, the "opposite" distribution with the opposite ground truth. Thus in this case, because we have assumed usefulness on these two distributions (and not just D), the assumption is different and stronger than Nagarajan and Kolter. The paragraph beginning "One subtlety" before Proposition 3.5 explained this deviation from Nagarajan and Kolter. We additionally justified this stronger assumption in footnote 3. We have revised to further clarify the text before Prop 3.5 to make it clearer that we are strengthening the assumption of NK.
> - **More direct proof of Theorem 3.1 and 3.2 via averaging?** Such an approach to study $\sum x_i y_i$ could work in the linear case, but we do not take this route, since it does not extend to nonlinear settings. Eg. in the XOR problem we study, such an approach would yield $E[x_i y_i]$= 0. Thus we study the max-margin solution since it can generalize both broadly.
> Further, in the linear problem, if $d = \omega(n)$, which is the regime in which we obtain a test loss of $o(1)$, then the set of $\epsilon$-max margin solutions contains the averaging solution $\sum x_i y_i$, but is a much larger family of solutions.

---

> > ### Author Response · Authors · 2022-11-11
> > **Response to Review (Part 2) - Clarity**
> >
> > **Clarity**
> >   - **Confusing assumptions in theorems.** There are two conditions in the all of the theorems:
> >      - a condition on the region of $\kappa$ (ie. $\kappa > \kappa_{gen}$ or $\kappa < \kappa_{uc}$)
> >      - An overparameterization condition requiring that d is large enough relative to n.
> >
> >     To increase clarity, we have revised our manuscript to include an additional more detailed version of Figure 1 with a 2-dimensional plot showing the regions where all of our results hold. This plot (Figure 2(right) in our revision) portrays both the conditions on $\kappa$ and the overparameterization assumption. We have also emphasized these two conditions in a revised first paragraph of Section 3.
> >     We note that the overparameterization condition for all of our neural net results is quite simple: we require that d > cn for some constant c. We emphasized this assumption in the Preliminaries, under the heading "Large Dimension Assumption".
> >     We agree that in the linear case (Theorems 3.1 and 3.5), the validity region of our results are harder to parse because of the technical overparameterization conditions pointed out in the reviewer’s comment. We have revised this to be more clear (see (1) below), and we explain in (2) below the reason for the technical assumptions, which we revise to move to remarks or the appendix.
> >       1. To simplify the exposition, we have modified these theorems to assume $\kappa$ is bounded away from the boundaries $\kappa_{gen}$ or $\kappa_{uc}$ by $\delta$ as in the neural net results (Thrm 3.2 and Prop 3.6). This simplifies the picture such that all of the relevant conditions for all theorems are clearly pictured in the new Figure 2(right).
> >       2.  We intentionally wanted our linear results to be strong enough to capture the subtle case when $\kappa \rightarrow 0$ (instead of being bounded away from 0 by a fixed $\delta$). Indeed, this regime is interesting because we can show that all classical margin bounds are _completely vacuous_, and it is also a regime of interest in the literature on linear classification, see eg. Shamir et al. 2022. Thus to allow for the biggest possible regime of $n, \sigma$, and $d$ in this subtle regime, we expressed the overparameterization condition as a function of $\kappa$. We now move these technical conditions (which yield tighter and stronger results) to the appendix.
> >   - **n needs to be small enough.** We discuss this under the heading "Large Dimension assumption" on page 4 in Preliminaries. Such an assumption that $d > n$ is standard in the literature on that goes beyond UC, and our assumption on the size of $d$ relative to $n$ is in fact mild relative to related work. [See the discussion in the paper under that heading].
> >   - **General technicality of Section 3.** We believe that Figure 1 (and now Figure 2(right)) and the paragraph at the beginning of Section 3 present the broad ideas of the theorems. The technicality of the theorems is largely due to the fact that we prove sharp thresholds for where generalization exists and UC fail. This type of tight thresholds requires some analysis terminology to convey.
> >   - **Is the extension to ($1-\epsilon$) maximum margin non-trivial?** The $(1-\epsilon)$ can indeed be derived from continuity for some epsilon. However, such an $\epsilon$ could be arbitrarily small, depending on $n$ or $d$. The non-triviality of this extension of our results comes from the fact that the $\epsilon$ is a constant, which does not depend on $n$ and $d$.
> > Note, technically $\epsilon$ depends on $\delta$ in the theorems, the boundary between $\kappa$ and the boundaries $\kappa_{gen}$ or $\kappa_{uc}$. However, one can treat this $\delta$ as a universal constant.
> >
> > Shamir et al. 2022 https://arxiv.org/abs/2201.11489

---

> > ### Author Response · Authors · 2022-11-11
> > **Response to Review (Part 3) - Reproducibility**
> >
> > **Reproducibility**
> >
> >   - **Confusion in Proof of Lemma B.1:** The fact that $v=\Xi(\Xi^T\Xi)^{-1}c$, comes from the fact that the min-norm solution to a linear regression problem can be written in closed form: $\text{argmin}_{X^Ta = b}\|a\|_2=X(X^TX)^{\dagger}b$
> > where $\dagger$ denotes the pseudo-inverse (see eg. Bartlett et al. 2020, page 5). Since $\Xi^T\Xi$ is invertible, the pseudo-inverse equals the inverse. We will include appropriate references and details on the pseudo-inverse calculation in a revision. The equation written by the reviewer is incorrect since the $d \times d$ matrix $\Xi\Xi^T$ is not necessarily invertible.
> >
> >     We believe our explanation that $\|v\|$ is in the specified interval is clear. In the last step of B.5, we have used conclusion (1) of the lemma (already proved), which states that $ \|\frac{1}{d\sigma^2}\Xi^T\Xi - I\|_2 < C\sqrt{n}{d}$. This is equivalent to saying the eigenvalues of $\Xi^T\Xi$ are in the range $d\sigma^2(1 - C\sqrt{n}{d}, 1 + \sqrt{n}{d})$.
> >
> >
> >   - **Request for detail in proof of Lemma C.1**: We have revised to include more detail in the proof. We use $c := \Xi^Tw$, where $w$ is the min-norm vector satisfying $w^T\xi_jy_j \geq \gamma$, where $\gamma := \min_j v^T\xi_jy_j$. We also use the fact that $\gamma$ must be positive otherwise the result is immediate.
> >   - **Typos in Proof of Lemma C.3 and on Page 7**: Revised, thank you!
> >   - **Confusion in Proof of Lemma A.1** We noticed and revised a typo where a $\sqrt{}$ around the $d-1$ is missing in the paragraph preceding and explaining this equation. [It should be $|f_v(x) = |v|\sigma|X|\sqrt{d-1}/\sqrt{X^2 + Y}$].
> > To reiterate the argument, if neither of the failure options occur, that is, (1) $X^2 + Y > d/4$ and (2) $|v|\sigma|X| \geq yf_u(x)/2$, then we have $|f_v(x) =  |v|\sigma|X|\sqrt{d-1}/\sqrt{X^2 + Y} \leq yf_u(x)\sqrt{d}$. Please let us know if there are any remaining confusions.
> > Proof of Theorem 3.1 uses $d>n$: The overparameterization assumption in the theorem ($d/n > c \max(1/\kappa^2, \kappa^2)$ implies that $d > n$ this so long as $c$ (in the theorem statement) is greater than 1. We can assume this since $c$ can be any universal constant.
> >
> > Bartlett et al. 2020 https://arxiv.org/pdf/1906.11300.pdf

---

> > ### Comment · Reviewer_DBpB · 2022-11-21
> > **Response to authors**
> >
> > Thank you for your reply. I looked at the updated paper and it is more clear now, particularly in its relation to Nagarjan and Kolter.
> >
> > Further, the points listed under "*Regarding Reproducibility*" were indeed just due to minor mistakes or missing details.
> >
> > There are two points I would like you to adress further.
> >
> > - Validity region of the results. I really like Figure 2, but it only applies to the linear setting. The XOR theorems make an additional assumption on the sample size, $n>c$. Either explicitly (Theorems 3.4 and 3.6) or implicitly (Theorems 3.2 and 3.3). With implicit I mean, that the bounds from these theorems hold with probability $1-3\exp(-n/c)$, which is $<0$ for $n\le c$. Could you add another figure for the validity region of the XOR problem in the $(n,d)$-plane. (A 3D plot including $\sigma$ would be great, but probably the plot is too convoluted, so 2D for fixed $\sigma$ is sufficient.)
> >
> > - Regarding the "large dimension assumption" and the sample size needing to be small enough. The paragraph on page 4 presents related work and their assumptions required for deriving generalization bounds, but that was not the point I was trying to make.
> > If we have a generalization bound which holds for a certain sample size, I would expect the bound to get a better (or at least not worse) when increasing the sample size. So far, theorems 3.1 and 3.2 have this behavior, as long as the sample size is not *too large*.
> > But what happens, if the sample size gets too large? For example, let $n_\max=c/d$ be the maximal allowed sample size. Do we have an upper bound on $\mathcal L_{\mathcal D}(f_w,n_\max+1)$? If yes, what would that be? Would this bound be better or worse than the bound on $\mathcal L_{\mathcal D}(f_w,n_\max)$.
> > If your results do not allow to estimate $\mathcal L_{\mathcal D}(f_w,n_\max+1)$, what would be the cause? It does not seems to be inherent to the data distribution, because $\mu$ can be estimated (as the class wise mean of the data) for every $n$, and this estimate improves with $n$. (Similarly, in the XOR problem $\mu_1, \mu_2$ can be estimated  as the mean of class wise differences). So I suspect that this limitation of the results would be because of the proof technique, or the level of detail in the inequalities.
> > To be clear, *if* the results do not allow to estimate $\mathcal L_{\mathcal D}(f_w,n_\max+1)$ and this is because of the proof technique, I do not consider it as a major flaw, as long as the root cause is identified and discussed, so that future work can built upon it.

---

> > > ### Author Response · Authors · 2022-11-23
> > > **Response on further two points**
> > >
> > > Thank you for following up and looking at the revision.
> > >
> > >
> > > **With regard to your first point on $n > c$:** We will revise with an additional figure alongside 2c showing the region of validity and good generalization for fixed $\sigma$ and variable $n$ and $d$.
> > >
> > > **Regarding increasing $n$:** We agree that this is a natural question and we will revise to include a discussion of this and revisions, as follows:
> > >
> > > When $n$ approaches $d$, the random matrix $\Xi$ becomes closer to square, and thus its smallest singular value becomes very small*, causing the eigenvalues of $(\Xi^T\Xi)^{-1}$ to explode, which means that Lemma B.1 cannot hold.
> > >
> > > In the linear problem, with a slight modification to the proof (replacing Lemma B.1 with a lemma which only shows an *upper bound* on the norm of $v$), we can show that our generalization results hold with no upper bound on $n$. In the XOR problem, because our proof uses Lemma B.1 in a more involved way, we do not see a quick way to revise the generalization results; however, we hypothesize that this is merely an artifact of the proof.
> > >
> > > Further, the explosion of the eigenvalues of $(\Xi^T\Xi)^{-1}$ means that any overfitting component must have a very large norm. This behavior means that (in the language of the “Key Inutitions” section), the “good” solution will have a better margin than the “bad” solution, and thus UC may actually hold for the max-margin solution, even above $\kappa_{uc}$.
> > >
> > > *See eg. https://arxiv.org/abs/0802.3956 for a discussion of the smallest singular value of random matrices.

---

> > > > ### Comment · Reviewer_DBpB · 2022-11-23
> > > > **Final response**
> > > >
> > > > Thank you for the explanation on $n$. I encourage you to include it in the paper.
> > > >
> > > > The key points of my review have been addressed and I will increase the rating.

---

### Decision · Program_Chairs · 2023-01-20

**Decision:**

Accept: poster

**Justification For Why Not Higher Score:**

The paper extends prior results of Nagarajan & Kolter. It is also high technical so probably of less interest to a wider audience.

**Justification For Why Not Lower Score:**

All reviewers agree this is worth accepting.

**Metareview: Summary, Strengths And Weaknesses:**

The paper derives novel generalization bounds in settings where uniform convergence provably
fails. More specifically, the new bounds derived in the paper are maximum margin bounds that can still generalize for linear and two-layer network models on distributions where uniform convergence fails.

The reviewers are overall positive. They did find the paper very technical and hard to read at times. One reviewer in particular was more negative before the discussion with the authors but agreed to increase their score afterward. I will recommend acceptance but I strongly urge the authors to revise the paper to make its results more accessible to a wider audience (for instance by providing further explanation of the key results and proof techniques).

**Note From Pc:**

if the above contains the word "oral" or "spotlight" please see: "oral" presentation means -> notable-top-5% and "spotlight" means -> notable-top-25%. As stated in our emails, we are disassociating presentation type from AC recommendations

**Summary Of Ac-Reviewer Meeting:**

N/A